# RESAMPLING GRADIENTS VANISH IN DIFFEREN-TIABLE SEQUENTIAL MONTE CARLO SAMPLERS

**Johannes Zenn**
University of Tübingen & IMPRS-IS
johannes.zenn@uni-tuebingen.de

**Robert Bamler**
University of Tübingen
robert.bamler@uni-tuebingen.de

## ABSTRACT

Annealed Importance Sampling (AIS) moves particles along a Markov chain from a tractable initial distribution to an intractable target distribution. The recently proposed Differentiable AIS (DAIS) (Geffner & Domke, 2021; Zhang et al., 2021) enables efficient optimization of the transition kernels of AIS and of the distributions. However, we observe a low effective sample size in DAIS, indicating degenerate distributions. We thus propose to extend DAIS by a resampling step inspired by Sequential Monte Carlo. Surprisingly, we find empirically—and can explain theoretically—that it is not necessary to differentiate through the resampling step, which avoids gradient variance issues observed in similar approaches for Particle Filters (Maddison et al., 2017a; Naesseth et al., 2018; Le et al., 2018).

## 1 SEQUENTIAL MONTE CARLO METHODS AND RESAMPLING

Sequential Monte Carlo (SMC) (Doucet et al., 2001; Liu & Liu, 2001) and Annealed Importance Sampling (AIS) (Neal, 2001)) are related methods to sample from unnormalized distributions and to estimate their normalization constants. SMC simulates the evolution of a set of particles through a Markov chain, where each transition takes three steps: (i) independent transitions of particles using either a given dynamics model (Particle Filters (PFs) (Gordon et al., 1993; Kong et al., 1994)) or Markov Chain Monte Carlo moves that interpolate between an initial and target distribution (SMC Samplers (Del Moral et al., 2006)); (ii) re-weighting each particle's probability; and (iii) resampling, i.e., replacing particles with low weights by "clones" of particles with high weights.

**Related Work** Differentiable PFs construct a lower bound on the log marginal likelihood utilizing the filtering distribution (e.g. Maddison et al. (2017a)) or the smoothing distribution (Lawson et al., 2022) of the PF as target distribution in each transition. Lawson et al. (2022) make this bound asymptotically tight. The recently proposed Differentiable AIS (DAIS) (Geffner & Domke, 2021; Zhang et al., 2021) can be phrased as an SMC Sampler without resampling (adapting its bound). DAIS omits the non-differentiable Metropolis-Hastings (MH) accept/reject step (Metropolis et al., 1953; Hastings, 1970) resulting in a differentiable method.

**Contributions** In high dimensions, DAIS suffers from a small Effective Sample Size (ESS) (Liu & Chen, 1998), i.e., a highly skewed distribution of particle weights where only few particles contribute (Figure 1, ✕). To overcome this limitation, we introduce resampling as in PFs and SMC Samplers. We optimize a Monte Carlo Objective (Mnih & Rezende, 2016) similar to the bound in PFs (Maddison et al., 2017a; Naesseth et al., 2018; Le et al., 2018) but adapted to SMC Samplers. By omitting the MH step (similar to DAIS), we obtain a Differentiable SMC Sampler (DSMCS, proposed).

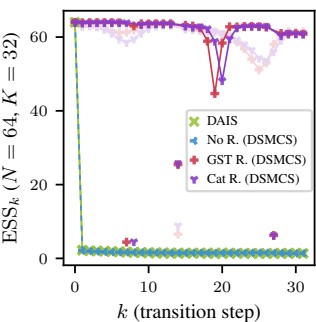

Figure 1: ESS for DAIS and DSMCS at epochs 100 and 500 (increasing opacity).

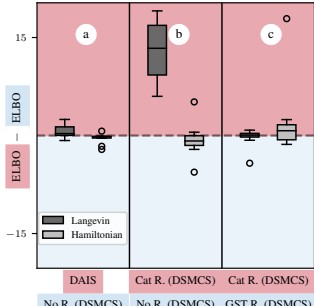

Figure 2: Difference of EL-BOs. Red and blue backgrounds show method with larger ELBO (red: top & upper row of x-axis labels).

Figure 2 summarizes differences of the Evidence Lower Bound (ELBO) between models with Langevin and Hamiltonian Markov kernel across a wide range of settings in a standard experiment (details in Appendix C). Subplots (a)-(c) build step by step to our main result that the proposed method (detailed in Section 2 below) does not require taking gradients due to resampling into account. In a first step, Figure 2 (a) compares DAIS to optimizing the proposed bound (Eq. (1) below) without resampling ("No R."). Both bounds are similarly tight. Figure 2 (b) introduces resampling but ignores resampling gradients ("Cat R."), which tightens the bound significantly for the Langevin kernel (additional results in Appendix D.1). Surprisingly, taking gradients due to resampling into account ("GST R.", Figure 2 (c))—which we do with the Gapped Straight Through (GST) estimator (Fan et al., 2022)—hardly affects the tightness of the bound. The next section explains this finding.

## 2 GRADIENTS DUE TO RESAMPLING IN A DIFFERENTIABLE SMC SAMPLER

We formalize DSMCS and its resampling gradient. We seek the normalization constant $Z := \int_{\mathbb{R}^n} \gamma(\boldsymbol{x}) \, \mathrm{d}\boldsymbol{x}$ of an unnormalized density $\gamma$ on $\mathbb{R}^n$. An SMC Sampler samples $N$ trajectories through a series $(\pi_k)_{k=0}^K$ of distributions $\pi_k(\boldsymbol{x}) \propto \gamma_k(\boldsymbol{x}) := \pi_0(\boldsymbol{x})^{1-\beta_k} \gamma(\boldsymbol{x})^{\beta_k}$ (with $0 = \beta_0 < \beta_1 < \ldots < \beta_K = 1$), which interpolate between a tractable initial distribution $\pi_0$ and the target $\pi_K = \gamma/Z$.

We describe an SMC Sampler algorithmically following Del Moral et al. (2006). First, one draws $N$ independent samples $\{\check{\boldsymbol{z}}_0^i\}_{i=1}^N$ from $\pi_0$. For each subsequent step $k \in \{1, \ldots, K\}$, one then draws samples $\{\boldsymbol{z}_k^i\}_{i=1}^N$ from a forward kernel $F_k(\boldsymbol{z}_k^i \mid \check{\boldsymbol{z}}_{k-1}^i)$ and one calculates importance weights $\tilde{w}_k(\boldsymbol{z}_k^i, \check{\boldsymbol{z}}_{k-1}^i) := \gamma_k(\boldsymbol{z}_k^i) B_k(\check{\boldsymbol{z}}_{k-1}^i \mid \boldsymbol{z}_k^i) / (\gamma_{k-1}(\check{\boldsymbol{z}}_{k-1}^i) F_k(\boldsymbol{z}_k^i \mid \check{\boldsymbol{z}}_{k-1}^i))$, where both Markov kernels $F_k$ and $B_k$ leave $\pi_k$ invariant. Finally, one (optionally) resamples from within the set of particles by setting $\check{\boldsymbol{z}}_k^i := \boldsymbol{z}_k^{\iota_k(i)}$ where the mapping $\iota_k : \{1, \ldots, N\} \to \{1, \ldots, N\}$ is drawn from a distribution $\mathcal{M}(\iota_k \mid \{w_k^i\}_{i=1}^N)$ that satisfies that each $\boldsymbol{z}_k^i$ is sampled $Nw_k^i$ times in expectation (i.e., $\mathbb{E}_{\mathcal{M}}\big[|\iota_k^{-1}(j)|\big] = Nw_k^i$), where $w_k^i = w_{k-1}^i \tilde{w}_k(\boldsymbol{z}_k^i, \check{\boldsymbol{z}}_{k-1}^i) / \sum_{j=1}^N (w_{k-1}^j \tilde{w}_k(\boldsymbol{z}_k^j, \check{\boldsymbol{z}}_{k-1}^j))$ and $w_0^i = 1/N$. In practice, the simplest way to satisfy this requirement is to draw the function values $\iota_k(i)$ i.i.d. for all $i$ from a categorical distribution with probabilities $(w_k^1, \ldots, w_k^N)$.

Similar to DAIS, DSMCS makes the Markov kernels $F_k$ and $B_k$ differentiable by not including a MH step. We adapt the bound proposed in the context of PFs (Maddison et al., 2017a; Naesseth et al., 2018; Le et al., 2018) to the setting of SMC Samplers and arrive at $\mathcal{L} < \log Z$ with

$$\mathcal{L} = \mathbb{E}_{[\prod_i \pi_0(\boldsymbol{z}_0^i)] \prod_{k=1}^K [\prod_i F(\boldsymbol{z}_k^i \mid \boldsymbol{z}_{k-1}^{\iota_{k-1}(i)})] \, \mathcal{M}(\iota_k \mid \{w_k^i\}_{i=1}^N)} \left[ \sum_{k=1}^K \log \left( \sum_{i=1}^N \alpha_k^i \, \tilde{w}_k(\boldsymbol{z}_k^i, \boldsymbol{z}_{k-1}^{\iota_{k-1}(i)}) \right) \right], \quad (1)$$

where the expectation $\mathbb{E}[\,\cdot\,]$ is taken over the sampling process described above, $\alpha_k^i = \frac{1}{N}$ if one resamples in step $k$ and $\alpha_k^i = w_k^i$ otherwise, and $\iota_0 = \mathrm{Identity}$ as resampling for $k = 0$ is not useful.

As the resampling distribution is discrete, differentiating through samples from it typically leads to either high gradient variance (e.g., when using REINFORCE (Glynn, 1990; Williams, 1992) gradient estimates) or to biased gradients (e.g., when using Gapped Straight Through (Fan et al., 2022) gradient estimates). However, the results in Figure 2 (c) indicate that differentiating through the resampling step is not necessary. This can be observed for models that resample in every transition (see Appendix D.1) and for models that resample with a probability (inversely) proportional to the ESS in a transition (Figure 2, (c); details in Appendix D.1). Additionally, we find that resampling tends to achieve high Effective Sample Sizes $\mathrm{ESS}_k := 1/\sum_{i=1}^N (w_k^i)^2 \in [1, N]$, regardless of whether we take gradients due to resampling into account (Figure 1, ✚) or not (Figure 1, ▼). It turns out that the high ESS indeed explains why resampling gradients are not necessary:

**Theorem 1** *The gradient of the resampling step vanishes if the ESS is maximal, i.e., if $ESS_k = N \, \forall k$.*
*Proof.* We prove the statement in Appendix A.

**Discussion** In practice, $\mathrm{ESS}_k$ is not maximal ($\mathrm{ESS}_k \approx N$ in Figure 1) and Theorem 1 may not hold. However, we can empirically verify that the gradients corresponding to the resampling operation indeed do not seem to have a significant impact on the results (see Figure 2). The work of Lawson et al. (2022) necessarily fulfills a maximal $\mathrm{ESS}_k$ (as it asymptotically makes the PF bound tight) which might explain why resampling gradients are not needed. Notably, leaving out resampling gradients reduces gradient variance that has been reported to hurt the estimation of the log normalization constant in PFs (Maddison et al., 2017a; Naesseth et al., 2018; Le et al., 2018).

URM STATEMENT

The corresponding author for this work meets the URM criteria of ICLR 2023 Tiny Papers Track being under-represented by age and being a first time submitter.

ACKNOWLEDGEMENTS

The authors would like to thank Tim Z. Xiao for helpful discussions. The authors thank the anonymous reviewers for their insightful comments and feedback that significantly improved the clarity of this paper. The authors would like to acknowledge support of the 'Training Center Machine Learning, Tübingen' with grant number 01—S17054. Funded by the Deutsche Forschungsgemeinschaft (DFG, German Research Foundation) under Germany's Excellence Strategy – EXC number 2064/1 – Project number 390727645. This work was supported by the German Federal Ministry of Education and Research (BMBF): Tübingen AI Center, FKZ: 01IS18039A. The authors thank the International Max Planck Research School for Intelligent Systems (IMPRS-IS) for supporting Johannes Zenn. Robert Bamler acknowledges funding by the German Research Foundation (DFG) for project 448588364 of the Emmy Noether Programme.

REPRODUCIBILITY STATEMENT

We release the code to reproduce all experiments at https://github.com/jzenn/DSMCS.

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

## A   PROOF OF THEOREM 1

We replicate Theorem 1 below and provide a proof.

**Theorem 1** *The gradient of the resampling step vanishes if the ESS is maximal, i.e., if $ESS_k = N \, \forall k$.*

*Proof.* We rewrite the expectation over $\{\iota_k\}_{k=1}^K$ in Eq. (1) as $\sum_{\{\iota_k\}_{k=1}^K} (\prod_k \mathcal{M}(\iota_k \,|\, \theta_k))[\cdots]$ where $\theta_k$ are opaque parameters that allow us to track gradients through $\mathcal{M}$. For each term in this sum, we trace back the indices $\{j_k\}_{k=0}^K$ that lead to index $j_K := 1$ at the last step $K$ by recursively defining $j_{k-1} := \iota_{k-1}(j_k)$. By Jensen's inequality for the strictly convex function $x \mapsto x^2$, an ESS of $N$ implies $w_k^i = \frac{1}{N} \, \forall k, i$, and thus $\tilde{w}_k(z_k^i, \check{z}_{k-1}^i) = \tilde{w}_k(z_k^i, z_{k-1}^{\iota_{k-1}(i)})$ is independent of $i$ for all $k$. We can thus replace the average over $i$ inside the logarithm in Eq. (1) with the single term with $i = j_k$,

$$\mathcal{L}_{(\text{ESS}=N)} = \sum_{\{\iota_k\}_{k=1}^K} \left( \prod_{k=1}^K \mathcal{M}(\iota_k \,|\, \theta_k) \right) \mathbb{E}_{\pi_0(z_0^{j_0}) \prod_{k=1}^K F(z_k^{j_k} \,|\, z_{k-1}^{j_{k-1}})} \left[ \sum_{k=1}^K \log \tilde{w}_k(z_k^{j_k}, z_{k-1}^{j_{k-1}}) \right]. \quad (2)$$

Here, we also marginalized trivially over all $z_k^i$ with $i \neq j_k$. The remaining coordinates $\{z_k^{j_k}\}_{k=0}^K$ describe a single trajectory that can no longer branch. Thus, they are *independent* integration variables, i.e., we may rename them by dropping the superscripts $j_k$ to stress that the expectation $\mathbb{E}[\cdot]$ over particle coordinates in Eq. (2) no longer depends on $\{\iota_k\}_{k=1}^K$. Thus, $\nabla_{\theta_k} \mathcal{L}_{(\text{ESS}=N)} = 0$.   □

## B  RELATED WORK

Figure 3 gives a brief overview of how the proposed DSMCS fits into the landscape of (differentiable) SMC and AIS methods.

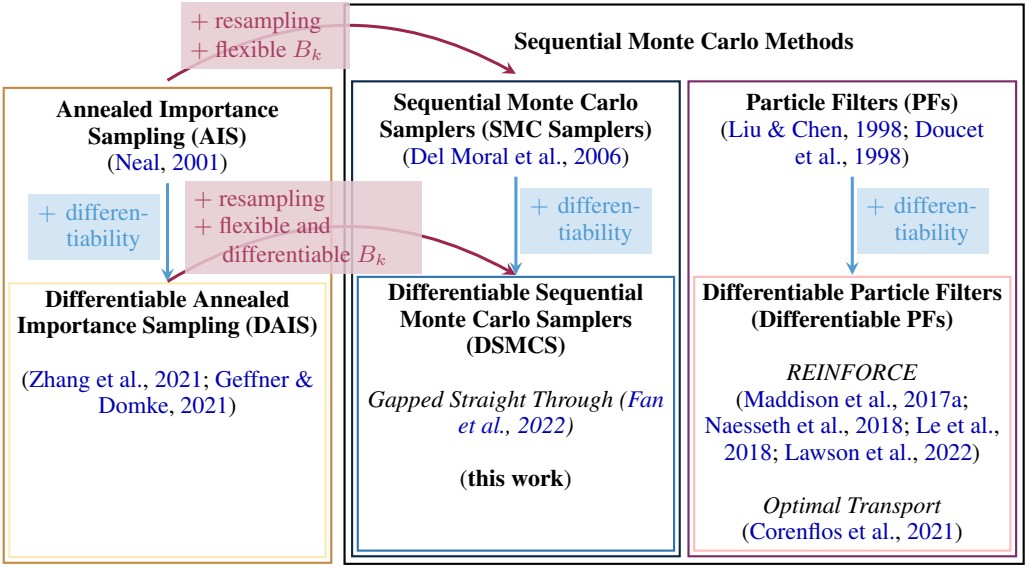

Figure 3: (Differentiable) AIS and SMC methods.

## C  DETAILS ON EXPERIMENTS

We replicate the static target experiment by Doucet et al. (2022). We estimate the log normalization constant of $p = \gamma/Z$ being a 50-dimensional Gaussian mixture with 8 components (note that $\log Z = 0$). The means of the Gaussian mixture $p$ are sampled from the Gaussian distribution $\mathcal{N}(3, \boldsymbol{I})$. The variance of each component is set to 1. The initial distribution $q = \pi_0$ is chosen to be a Gaussian with large diagonal variance, namely $\mathcal{N}(0, 9\boldsymbol{I})$. We run the DSMCS with various resampling schemes for $K \in \{8, 16, 32\}$ and $N \in \{8, 32, 64\}$ utilizing a small neural network $\mathrm{NN}(\cdot)$ predicting the step sizes $\{\delta_k = \hat{\delta} \cdot \sigma(\mathrm{NN}(k))\}_{k=1}^K$ with the same architecture as proposed by Doucet et al. (2022). Additionally, we learn the annealing schedule $\beta_k$. For the Hamiltonian Markov kernel we also learn the scale $c$ of the mass matrix $\boldsymbol{M} = c\boldsymbol{I}$ and a scalar for the momentum refreshment $\rho$. All models maximize the bound $\mathcal{L}$ for 500 epochs (each consisting of 10 iterations) with the Adam optimizer (Kingma & Ba, 2014) in batches of 64. We scale the learning rate every 25 epochs by a factor of 0.75 for the first 200 epochs.

Appendix C.1 gives an overview of the resampling schemes that we evaluate the DSMCS models with on the static target experiment. Appendix C.2 describes the unadjusted overdamped Langevin kernel utilized in our experiments. Appendix C.3 describes the Hamiltonian Markov kernel (that incorporates Hamiltonian dynamics via the underdamped Langevin equation). For a thorough derivation of these kernels, please see the original work of Doucet et al. (2022).

### C.1  MODELS

In the main text we only show results for the models **Bern-Cat** and **Bern-GST** to convey the main message but call the models **Cat** and **GST** to keep notation simple. In this supplement and all following ones, we call the models as they are described below.

We employ the proposed DSMCS with multiple resampling schemes that either include the resampling gradient or ignore it. The distribution $\mathcal{M}(\iota_k \mid \{w_k^i\}_{i=1}^N)$ takes the following four functional forms.

- **Cat** (Categorical Resampling (without gradients)): We choose a categorical distribution for $\mathcal{M}(\iota_k \mid \{w_k^i\}_{i=1}^N)$ and draw $\iota_k(i) \sim \mathrm{Cat}(\{w_k^i\}_{i=1}^N)$.

- **Bern-Cat** (Categorical Resampling (without gradients) with Bernoulli Relaxation (without gradients)): We augment the categorical distribution from above with an additional Bernoulli random variable that "decides" whether to resample at iteration $k$ for each batch, namely $\iota_k(i) = (1 - b_k)i + b_k c_k^i$, where $c_k^i \sim \mathrm{Cat}(\{w_k^i\}_{i=1}^N)$ and $b_k \sim \mathrm{Bern}\left(1 - (1/\sum_{j=1}^N (w_k^j)^2 - 1)/(N - 1)\right)$. This idea is inspired by SMC where one typically employs a (hard) threshold of $N/2$ and resamples particles if the threshold is undercut.

- **GST** (Gapped Straight Through Gumbel-Softmax Resampling (with gradients)): This resampling scheme is similar to **Cat** but with resampling gradients. We utilize the Gapped Straight Through (GST) estimator (Fan et al., 2022) for the categorical distribution. This estimator is known to reduce gradient variance compared to the Gumbel-Softmax (Jang et al., 2017; Maddison et al., 2017b) estimator. We train models with two temperatures $\tau \in \{0.1, 1.0\}$.

- **Bern-GST** (Gapped Straight Through Gumbel-Softmax Resampling with Bernoulli Relaxation (with gradients)): This resampling scheme builds on the **GST** resampling scheme but also reparametrizes the Bernoulli variable utilizing the GST estimator. We train models with two temperatures $\tau \in \{0.1, 1.0\}$.

## C.2 Langevin Markov Kernel

For a specific $k$ we choose the Markov kernels

$$F_k(z_k^i \mid \check{z}_{k-1}^i) = \mathcal{N}(z_k^i \mid \check{z}_{k-1}^i + \delta_k \nabla \log \gamma_k(\check{z}_{k-1}^i), 2\delta_k \boldsymbol{I}),$$
$$B_k(\check{z}_{k-1}^i \mid z_k^i) = F_k(\check{z}_{k-1}^i \mid z_k^i),$$

and compute the incremental weight as

$$\tilde{w}_k = \frac{\gamma_k(z_k^i)B_k(\check{z}_{k-1}^i \mid z_k^i)}{\gamma_{k-1}(\check{z}_{k-1}^i)F_k(z_k^i \mid \check{z}_{k-1}^i)},$$

where $\gamma_k(z_k^i) = (1 - \beta_k)\log q(z_k^i) + \beta_k \log p(z_k^i)$.

## C.3 Hamiltonian Markov Kernel

For the Hamiltonian kernel, let $v$ denote the momentum variable, let $\boldsymbol{M}$ denote the mass matrix, let $\rho$ denote the damping coefficient, and let $L(z_k^i, v_k^i)$ denote the leap frog integrator for variables $z_k^i, v_k^i$ utilizing a step size of $\delta_k$. We have

$$\tilde{v}_k^i \sim \mathcal{N}(\rho \check{v}_{k-1}^i, (1 - \rho^2)\boldsymbol{M}),$$
$$\text{and } z_k^i, v_k^i = L(\check{z}_{k-1}^i, \tilde{v}_k^i).$$

We compute the incremental weight as

$$\tilde{w}_k = \frac{\gamma_k(z_k, v_k)\mathcal{N}(\check{v}_{k-1}^i \mid \rho\tilde{v}_k^i, (1 - \rho^2)\boldsymbol{M})}{\gamma_{k-1}(\check{z}_{k-1}^i, \check{v}_{k-1}^i)\mathcal{N}(\tilde{v}_k^i \mid \rho\check{v}_k^i, (1 - \rho^2)\boldsymbol{M})},$$

where $\gamma_k(z_k^i, v_k^i) = (1 - \beta_k)\log q(z_k^i) + \beta_k \log p(z_k^i) + \log \mathcal{N}(v_k^i, \boldsymbol{0}, \boldsymbol{M})$.

## D Quantitative Results

Appendix D.1 discusses Figure 4 showing results of models with resampling and Bernoulli decision ("Bern-Cat R." and "Bern-GST R.") as well as models with resampling every iteration ("Cat R." and "GST R."). Appendix D.2 discusses Figure 1 of the main text in detail and provides quantitative results. Appendix D.3 discusses Figure 2 of the main text and provides quantitative results for all experiments.

### D.1 CATEGORICAL RESAMPLING WITH BERNOULLI DECISION

SMC methods typically resample only if the effective sample size drops below a threshold of $N/2$ where $N$ is the total number of particles. This prevents information from being lost and reduces the variance of the estimator. Thus, we introduce for the DSMCS a Bernoulli random variable that "decides" whether to resample at iteration $k$ for each batch, namely $\iota_k(i) = (1-b_k)i + b_k c_k^i$, where

$$c_k^i \sim \text{Cat}(\{w_k^i\}_{i=1}^N) \quad \text{and} \quad b_k \sim \text{Bern}\left(1 - \left(1 \Big/ \sum_{j=1}^N (w_k^j)^2 - 1\right) \Big/ (N-1)\right).$$

Appendix C.1 gives more details on the Bernoulli relaxed models.

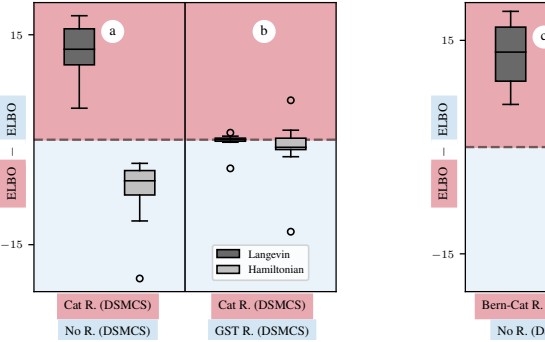

Figure 4: Difference of ELBOs for models without Bernoulli decision (left) and with Bernoulli decision (right). The right plot is copied from the main text (Figure 2 (b) & (c)). Red and blue backgrounds show the method achieving larger ELBO (red: top & upper row of x-axis labels).

We find that DSMCS employing categorical resampling without Bernoulli decision ("Cat R.") tightens the bound for the Langevin kernel (Figure 4 (a)) compared to DSMCS without resampling ("No R."). This finding is similar to the results in the main text for models with Bernoulli decision (Figure 4 (c)). Contrary to findings in the main text, we find a negative impact of performing resampling in DSMCS with the Hamiltonian kernel (Figure 4 (a) in contrast to Figure 4 (c)). This might be due to the increased variance and information loss of the model without Bernoulli decision. Additionally, we verify that taking gradients due to resampling into account ("GST R.") compared to not taking gradients into account ("Cat R.") hardly affects the tightness of the bound (see Figure 4 (b)). Also, this result is similar to the results described in the main text (see Figure 4 (d)).

### D.2 EFFECTIVE SAMPLE SIZE

Table 1 shows quantitative results of Figure 1. We visualize the ESS for epochs 100 and 500 with different opacities for experiments with the Langevin kernel. We visualize the DSMCS models without resampling ("No R.") and models with resampling that employ a Bernoulli decision for resampling with gradients ("GST R." with $\tau = 0.1$) and without gradients ("Cat R."). The ESS is an average of 10 iterations and the batch size of 64.

Table 1: Quantitative results of ESS for DAIS, No Resampling, Bernoulli Resampling ("Bern-GST R."), and Bernoulli categorical resampling ("Bern-Cat R."). Details in text.

| $k$ | 1 | 2 | 3 | 4 | 5 | 6 | 7 | 8 | 9 | 10 | 11 | 12 | 13 | 14 | 15 | 16 |
|---|---|---|---|---|---|---|---|---|---|---|---|---|---|---|---|---|
| DAIS | 64.0 | 2.18 | 2.06 | 2.0 | 1.97 | 1.91 | 1.85 | 1.78 | 1.72 | 1.67 | 1.63 | 1.59 | 1.55 | 1.54 | 1.51 | 1.49 |
| No Resampling | 64.0 | 2.09 | 1.95 | 1.86 | 1.77 | 1.7 | 1.68 | 1.65 | 1.62 | 1.61 | 1.59 | 1.57 | 1.53 | 1.49 | 1.48 | 1.48 |
| Bern-GST R. | 64.0 | 63.91 | 63.94 | 63.95 | 63.95 | 63.94 | 63.88 | 4.41 | 63.03 | 63.62 | 63.6 | 63.63 | 63.6 | 63.63 | 25.42 | 63.21 |
| Bern-Cat R. | 64.0 | 63.8 | 63.89 | 63.92 | 63.93 | 63.93 | 63.93 | 63.9 | 4.51 | 63.37 | 63.57 | 63.58 | 63.57 | 63.57 | 25.72 | 62.95 |

| $k$ | 17 | 18 | 19 | 20 | 21 | 22 | 23 | 24 | 25 | 26 | 27 | 28 | 29 | 30 | 31 | 32 |
|---|---|---|---|---|---|---|---|---|---|---|---|---|---|---|---|---|
| DAIS | 1.48 | 1.49 | 1.51 | 1.49 | 1.51 | 1.48 | 1.42 | 1.39 | 1.38 | 1.36 | 1.36 | 1.42 | 1.43 | 1.43 | 1.46 | 1.42 |
| No Resampling | 63.01 | 62.49 | 58.88 | 44.69 | 58.38 | 62.8 | 62.77 | 62.88 | 62.92 | 62.85 | 62.67 | 6.22 | 61.0 | 60.81 | 60.96 | 60.79 |
| Bern-GST R. | 60.62 | 8.18 | 62.44 | 61.98 | 62.44 | 62.46 | 62.42 | 62.47 | 62.33 | 6.1 | 59.37 | 59.94 | 59.55 | 60.04 | 60.08 | 63.4 |
| Bern-Cat R. | 62.65 | 62.79 | 62.27 | 58.85 | 48.19 | 59.51 | 62.49 | 62.78 | 62.95 | 62.95 | 62.88 | 6.26 | 60.38 | 61.08 | 60.86 | 60.9 |

We find that the ESS of DAIS and DSMCS without resampling drops after just one a iteration to value around 1.5. For the Bern-GST R. model (utilizing straight-through gradients corresponding to the resampling operation) we find that the ESS stays mostly at around values of 63, dropping for larger $k \geq 16$ to values around 60. The Bern-Cat R. model (trained without any gradients corresponding to resampling) shows similar behavior. For the models utilizing resampling (with and without resampling gradients) we also notice a few outliers.

## D.3 STATIC TARGETS

This section provides quantitative results utilized in this work. Table 2 shows results on the static target experiment of the Langevin Markov kernel and Table 3 shows results of the Hamiltonian Markov kernel for various $N$, $K$, resampling schemes, and step sizes. For each entry in the table we search over learning rates $10^{-2}, 3 \cdot 10^{-2}, 5 \cdot 10^{-2}, 9 \cdot 10^{-2}$. The model that achieves the best results is then trained on three different seeds.

Table 2: Results of Langevin kernel (means and standard deviations over three runs). For details see main text. N/A denotes experiments that did not converge. Appendix C describes the experimental setup. For a description of the models please consult Appendix C.1.

| $\delta$ | | DAIS | | | | No | | | | Cat | | | | Bern-Cat | | |
|---|---|---|---|---|---|---|---|---|---|---|---|---|---|---|---|---|
| | $K$ \ $N$ | 8 | 32 | 64 | $K$ \ $N$ | 8 | 32 | 64 | $K$ \ $N$ | 8 | 32 | 64 | $K$ \ $N$ | 8 | 32 | 64 |
| 0.1 | 8 | -174.3±1.93 | -146.89±6.21 | -141.63±0.5 | 8 | -177.97±4.79 | -149.09±2.65 | -144.92±2.94 | 8 | -172.21±2.36 | -141.7±3.15 | -129.57±1.11 | 8 | -173.67±5.13 | -140.68±4.54 | -127.16±2.52 |
| | 16 | -139.28±0.77 | -111.98±3.66 | -104.39±4.33 | 16 | -134.17±1.65 | -111.67±2.49 | -105.46±2.95 | 16 | -124.21±3.67 | -96.42±2.48 | -89.19±2.32 | 16 | -122.16±3.53 | -96.9±1.52 | -85.31±0.41 |
| | 32 | -84.27±2.7 | -73.13±0.92 | -65.89±0.92 | 32 | -86.27±2.66 | -75.56±2.4 | -68.46±0.83 | 32 | -75.55±2.2 | -63.31±1.94 | -59.62±3.0 | 32 | -76.06±2.35 | -62.33±0.71 | -59.9±1.65 |
| 0.25 | 8 | -115.54±0.56 | -96.27±4.5 | -87.98±4.22 | 8 | -116.89±2.47 | -97.59±1.96 | -88.04±1.44 | 8 | -103.96±1.35 | -80.89±2.19 | -72.66±0.76 | 8 | -103.77±3.36 | -79.73±1.42 | -71.18±1.72 |
| | 16 | -72.48±1.64 | -59.37±1.76 | -54.85±0.8 | 16 | -71.72±0.84 | -61.83±3.16 | -54.88±0.59 | 16 | -59.36±2.14 | -44.11±1.82 | -39.02±1.08 | 16 | -58.37±1.53 | -42.75±0.98 | -38.45±1.54 |
| | 32 | -40.19±1.3 | -33.14±0.58 | -29.27±1.09 | 32 | -40.48±0.88 | -32.58±0.23 | -29.69±1.22 | 32 | -29.78±0.98 | -26.9±0.53 | -25.18±0.81 | 32 | -31.22±0.79 | -26.59±0.07 | -23.16±0.95 |
| 1.0 | 8 | -31.79±0.25 | -25.92±1.13 | -24.33±0.36 | 8 | -32.11±0.5 | -25.71±1.78 | -23.17±0.7 | 8 | -23.88±0.61 | -15.53±0.49 | -12.74±0.36 | 8 | -24.32±0.7 | -15.79±0.53 | -13.31±0.35 |
| | 16 | -18.19±1.0 | -14.16±0.11 | -12.2±0.48 | 16 | -18.62±0.23 | -14.3±0.42 | -12.52±0.44 | 16 | -11.22±0.39 | -7.4±0.88 | -8.34±1.65 | 16 | -11.83±0.4 | -8.38±0.33 | -9.18±0.35 |
| | 32 | -9.88±0.29 | -7.12±0.25 | -5.93±0.08 | 32 | -9.89±0.47 | -6.84±0.23 | -5.97±0.27 | 32 | -6.87±0.07 | -16.59±0.7 | -22.96±7.68 | 32 | -7.62±0.48 | -4.11±0.55 | N/A |

| $\delta$ | | GST ($\tau = 1.0$) | | | | Bern-GST ($\tau = 1.0$) | | | | GST ($\tau = 0.1$) | | | | Bern-GST ($\tau = 0.1$) | | |
|---|---|---|---|---|---|---|---|---|---|---|---|---|---|---|---|---|
| | $K$ \ $N$ | 8 | 32 | 64 | $K$ \ $N$ | 8 | 32 | 64 | $K$ \ $N$ | 8 | 32 | 64 | $K$ \ $N$ | 8 | 32 | 64 |
| 0.1 | 8 | -190.2±6.68 | -151.38±1.99 | -148.92±10.88 | 8 | -183.73±5.01 | -145.47±2.18 | -136.43±13.44 | 8 | -172.3±2.42 | -141.67±3.04 | -130.08±1.12 | 8 | -173.84±4.92 | -139.26±2.95 | -126.92±2.85 |
| | 16 | -147.98±10.44 | -116.82±0.74 | -122.78±10.71 | 16 | -141.35±4.55 | -108.57±5.38 | -90.19±10.53 | 16 | -124.31±3.65 | -96.6±2.72 | -88.81±3.07 | 16 | -122.03±3.72 | -96.98±1.53 | -85.66±1.53 |
| | 32 | -103.87±9.24 | -69.95±8.46 | -50.74±6.98 | 32 | -76.51±2.85 | -50.43±0.3 | -38.86±0.3 | 32 | -75.64±2.27 | -62.72±1.97 | -62.69±0.76 | 32 | -74.28±2.78 | -61.28±0.86 | -60.49±0.9 |
| 0.25 | 8 | -105.61±4.14 | -79.78±2.3 | -72.53±0.79 | 8 | -109.65±6.27 | -79.06±1.84 | -69.29±1.43 | 8 | -104.0±1.65 | -80.54±1.34 | -73.17±0.58 | 8 | -103.84±3.32 | -79.59±1.25 | -70.46±2.09 |
| | 16 | -66.4±11.06 | -43.13±2.6 | -39.35±5.41 | 16 | -63.07±6.52 | -44.66±4.22 | -33.82±0.22 | 16 | -59.24±2.24 | -44.34±1.18 | -40.03±0.92 | 16 | -58.45±2.07 | -43.56±1.0 | -38.21±0.37 |
| | 32 | -32.37±1.44 | -17.98±0.87 | -14.18±0.46 | 32 | -31.09±1.13 | -15.53±0.6 | -11.13±0.4 | 32 | -29.84±0.89 | -26.69±0.72 | -21.09±4.08 | 32 | -32.04±0.82 | -26.92±0.27 | -18.93±1.36 |
| 1.0 | 8 | -31.26±6.48 | -19.78±0.42 | -15.41±0.73 | 8 | -25.75±3.15 | -19.27±2.79 | -12.17±0.39 | 8 | -23.89±0.41 | -15.69±0.73 | -12.76±0.74 | 8 | -24.04±0.84 | -15.77±0.15 | -13.49±0.48 |
| | 16 | -24.82±3.89 | -8.17±0.63 | -6.04±0.14 | 16 | -48.11±24.02 | -7.46±0.25 | -6.32±0.3 | 16 | -11.61±0.97 | -7.62±1.02 | -6.18±0.28 | 16 | -11.42±0.6 | -8.2±1.34 | -6.46±0.74 |
| | 32 | -22.55±15.4 | -6.88±2.11 | -5.97±3.66 | 32 | -16.05±5.7 | -4.21±0.13 | N/A | 32 | -6.66±0.17 | -16.39±0.28 | -18.75±0.34 | 32 | -7.57±0.16 | -4.05±0.22 | -6.3±2.16 |

Table 3: Results of Hamiltonian kernel (means and standard deviations over three runs). For details see main text. N/A denotes experiments that did not converge. Appendix C describes the experimental setup. For a description of the models please consult Appendix C.1.

| $\delta$ | | DAIS | | | | No | | | | Cat | | | | Bern-Cat | | |
|---|---|---|---|---|---|---|---|---|---|---|---|---|---|---|---|---|
| | $K$ \ $N$ | 8 | 32 | 64 | $K$ \ $N$ | 8 | 32 | 64 | $K$ \ $N$ | 8 | 32 | 64 | $K$ \ $N$ | 8 | 32 | 64 |
| 0.1 | 8 | -15.35±1.9 | -11.36±0.65 | -9.72±0.16 | 8 | -14.3±0.54 | -11.14±0.38 | -10.05±0.32 | 8 | -24.15±1.3 | -16.59±0.75 | -14.88±0.7 | 8 | -20.14±0.46 | -14.56±0.37 | -10.57±0.45 |
| | 16 | -9.22±0.73 | -7.34±0.59 | -5.79±0.47 | 16 | -9.49±0.79 | -7.58±0.27 | -6.11±0.53 | 16 | -14.67±0.58 | -11.44±0.64 | -11.35±2.93 | 16 | -11.06±0.89 | -6.21±0.28 | -10.83±4.83 |
| | 32 | -6.27±0.44 | -4.63±0.41 | -3.81±0.71 | 32 | -6.85±0.92 | -6.52±1.01 | -3.61±0.44 | 32 | -7.55±1.07 | -6.04±1.52 | -3.96±0.36 | 32 | -4.36±0.84 | -4.63±0.1 | -1.98±0.2 |
| 0.25 | 8 | -14.31±0.39 | -11.16±0.28 | -9.61±0.42 | 8 | -14.98±0.73 | -11.04±0.74 | -9.46±0.45 | 8 | -26.58±1.72 | -18.94±0.31 | -15.33±0.44 | 8 | -20.58±0.37 | -13.16±0.71 | -10.98±0.48 |
| | 16 | -9.46±0.93 | -6.18±0.22 | -5.43±0.21 | 16 | -9.08±0.41 | -5.97±0.23 | -5.23±0.02 | 16 | -13.49±0.22 | -8.95±0.49 | -8.95±0.49 | 16 | -9.48±0.56 | -6.8±0.29 | -5.26±1.03 |
| | 32 | -5.96±0.34 | -5.35±0.77 | -6.27±4.37 | 32 | -5.87±0.45 | -3.7±0.46 | -4.14±0.89 | 32 | -10.42±1.39 | -23.53±22.83 | -10.4±3.26 | 32 | -5.36±0.25 | -4.69±0.44 | 1.0±0.14 |
| 1.0 | 8 | -15.73±1.09 | -11.22±0.27 | -9.18±0.11 | 8 | -14.26±0.54 | -10.3±0.2 | -9.28±0.14 | 8 | -25.91±0.87 | -28.69±1.41 | -18.42±1.33 | 8 | -23.83±0.76 | -25.6±2.12 | -18.79±2.09 |
| | 16 | -8.57±0.89 | -6.4±0.45 | -5.9±0.91 | 16 | -9.04±1.03 | -5.52±0.35 | -6.6±1.29 | 16 | -18.23±2.54 | -13.76±1.2 | -15.38±3.86 | 16 | -14.04±1.17 | -8.1±1.84 | -11.64±5.23 |
| | 32 | -5.26±0.43 | -3.57±0.17 | -3.31±0.35 | 32 | -5.85±0.18 | -6.0±1.79 | -4.76±1.97 | 32 | -12.17±1.15 | -12.65±0.79 | -13.46±1.79 | 32 | N/A | -5.82±0.48 | -4.64±0.4 |

| $\delta$ | | GST ($\tau = 1.0$) | | | | Bern-GST ($\tau = 1.0$) | | | | GST ($\tau = 0.1$) | | | | Bern-GST ($\tau = 0.1$) | | |
|---|---|---|---|---|---|---|---|---|---|---|---|---|---|---|---|---|
| | $K$ \ $N$ | 8 | 32 | 64 | $K$ \ $N$ | 8 | 32 | 64 | $K$ \ $N$ | 8 | 32 | 64 | $K$ \ $N$ | 8 | 32 | 64 |
| 0.1 | 8 | -228.25±0.69 | -138.01±17.07 | -125.01±10.53 | 8 | -116.69±74.94 | -108.62±7.04 | -85.47±7.21 | 8 | -25.34±2.16 | -16.58±1.51 | -15.34±0.15 | 8 | -22.94±1.77 | -13.64±0.12 | -12.02±0.38 |
| | 16 | N/A | N/A | N/A | 16 | N/A | -199.13±10.01 | -181.45±7.17 | 16 | -15.88±0.29 | -11.01±1.15 | -11.82±5.66 | 16 | -18.27±6.09 | -7.77±1.93 | -8.54±0.5 |
| | 32 | N/A | N/A | -176.48±9.59 | 32 | N/A | -26.49±19.09 | -4.62±3.68 | 32 | -9.3±1.76 | -7.49±1.1 | -5.69±0.27 | 32 | -6.21±3.1 | -2.74±0.14 | -3.48±0.45 |
| 0.25 | 8 | -234.5±12.2 | -122.28±36.56 | -50.93±12.21 | 8 | -58.96±20.3 | -103.45±44.46 | -38.46±7.0 | 8 | -24.16±1.34 | -17.58±0.03 | -14.25±0.04 | 8 | -23.02±2.82 | -14.22±0.7 | -10.03±0.5 |
| | 16 | -156.89±34.08 | -150.69±42.62 | -104.45±16.28 | 16 | -99.05±59.42 | -101.3±6.31 | -128.49±48.3 | 16 | -13.35±0.19 | -9.57±1.15 | -14.59±5.35 | 16 | -9.89±0.72 | -24.69±4.25 | -4.71±0.73 |
| | 32 | -59.33±28.63 | -36.89±17.48 | -32.68±14.01 | 32 | N/A | -4.93±4.57 | 0.02±0.96 | 32 | -9.01±0.53 | -10.39±1.85 | -11.76±1.24 | 32 | -4.0±0.36 | N/A | -0.39±0.55 |
| 1.0 | 8 | -35.55±2.63 | -38.06±8.11 | -44.11±15.84 | 8 | -30.82±3.26 | -31.95±5.74 | -49.46±9.09 | 8 | -25.95±1.72 | -22.38±1.98 | -20.66±2.85 | 8 | -24.79±2.27 | -17.22±0.31 | -15.86±0.55 |
| | 16 | -25.94±6.86 | -29.06±7.48 | -45.72±7.83 | 16 | -13.74±1.14 | N/A | -47.35±20.14 | 16 | -15.63±0.62 | -14.62±1.86 | -37.45±12.66 | 16 | N/A | -8.29±2.11 | N/A |
| | 32 | -11.34±2.02 | -244.71±296.56 | -17.58±1.35 | 32 | N/A | N/A | -5.16±0.12 | 32 | -11.81±2.21 | -16.4±5.07 | -19.54±4.19 | 32 | N/A | -8.88±2.08 | N/A |

