# OpenReview forum: "Resampling Gradients Vanish in Differentiable Sequential Monte Carlo Samplers"
_ICLR.cc/2023/TinyPapers — Submitted to Tiny Papers @ ICLR 2023_

### Official Review · Reviewer_6Bx1 · 2023-03-26

**Confidence:** 4

**Summary Of Contributions:**

This paper proposes using a resampling operation in (D)AIS, instead of prior methods which use a non-differentiable MH update step or a learned kernel.  The authors provide evidence that resampling achieves higher ESSs (with a similar feel to how SMC improves over regular IS) on an existing benchmark problem.  Some theoretical analysis is also provided.

**Rating:**

High Potential (HP): a submission which meets the reviewing criteria and has potential to make an impact on the field

**Strengths And Weaknesses:**

This is a strong submission, which actually has the feel of a full-length paper that has been aggressively cut down to length.  I query if this is the intention of TinyPapers as a venue (but the work itself cannot be faulted on that front).  There have been several recent papers on differentiable SMC/AIS-like algorithms (DAIS, SIXO), and so this work would be a nice addition to the literature.

## Questions
1. The authors' methods achieve nearly perfect ESS, which, if true, is very compelling.  I query how the ESS is being computed.  The ESS should be computed from the weights prior to resampling.  It isn’t fully surprising that the methods with resampling get higher ESS, because the particles’ weights are reset after resampling.  A fully degenerate SMC sampler (where just one particle survives) can achieve a near-perfect ESS after resampling if a single particle is iterated into a flat region of the joint probability (all iterated particles are very similar and hence equally weighted).  I invite the authors to comment on this, clarify how they compute the ESS, and maybe provide additional context on how particles differ.  Why is the ESS for “No Resampling” so low at the start?  Should the ESS for all methods at step zero be approximately the same?
2. I think theorem 2 is trivially correct?  If the ESS = N, then there is effectively no resampling, or rather, since the particles are all equally likely, then there is no preferential signal over the “wiring” of particles.  I think this is okay, it is always nice to write out foundational results, but this theorem could be dropped to the supplement to free space for more discussion.

I think the result in theorem 1 is however a nice parallel to the follow-up work to Maddison et al. (2017a), SIXO (Lawson et al., 2022), where the backwards message asymptotically makes the bound tight (and hence all particles are equally weighted and ESS = N).  Drawing an explicit parallel (in the discussion section) between the two works would be a nice way to link in to some more recent literature as well.

*Slight edit from my original review*:  I think proving out that resampling gradients go to zero in the limit of perfect ESS is worthwhile, but it is especially important to explicitly link this to the fact that your experiments seem to recover near-perfect ESS.  I would like to see an additional comment/caveat that this proof (and associated benefit) only holds in an asymptotic range.  On more complex examples, where perfect ESS is not recovered, it doesn't apply, and you fall back to the empirical evidence of the benefit, as per Maddison.

## Clarity

The paper itself is pretty well written and reads clearly to someone who is familiar with the topic.  The paper is very condensed and is intense technical for such a short paper, which may put some readers off, but, again, may be the result of the format.

My main concern with the clarity is that I really struggle to parse the results.  Figure 1 is not super clear.  I think “No Resampling” is regular (D)AIS, and “GST” and “CAT” are two variants of the authors work? I am to take from this that no resampling is always almost entirely degenerate?

I then have no idea what Figure 2 is showing.  The colors, markers, dashed lines etc are all very confusing.  For the main paper, I recommend thinning the results and focusing on just one of Langevin or Hamiltonian (include the other in the appendix), and really highlighting whatever the main result is.

## Correctness
I think the work is correct, modulo some clarification w.r.t. ESS above.

## Reproducibility
The authors have provided a lot of additional detail in the supplement.  I believe the work would be reproducible.


**Suggested Changes:**

See above in “Clarity”.  My main request is “thinning” the results figures, and highlighting just a single positive result in the main text.  The more complete results can then be deferred to the supplement.

With this distillation of *the* key result, drawing explicit connections to Lawson et al., and clarifying how ESS is computed/expanding on the weaknesses of ESS as a metric, I would be comfortable to rating this paper even higher.

Good work, and good luck.

---

> ### Author Response · Authors · 2023-04-20
> **Response to Reviewer 6Bx1**
>
> We would like to thank you very much for your thorough review and the time you invested. The review raised important questions and resulted in a significantly improved paper.
>
> >This is a strong submission, which actually has the feel of a full-length paper that has been aggressively cut down to length. [...] There have been several recent papers on differentiable SMC/AIS-like algorithms (DAIS, SIXO), and so this work would be a nice addition to the literature.
>
> Thank you very much for your kind summary. We chose the DEI initiative as the main author fulfills two criteria. We also think that the new “Tiny Paper” format is ideal for presenting these kinds of small results that can be summarized briefly and that can potentially make other researchers work easier, and which might get buried when presented as part of a longer paper.
> We see that the paper is written very densely. We uploaded a new version that improves on several points:
>
> - The paper now contains a section on “Related Work” pulling together all parts discussing how the paper fits into the current research, and a section “Contributions” that summarizes the novelty.
> - The paper only mentions one set of models (No R., Cat, GST), pushing details and additional results to the appendix.
> - The text explicitly conveys the main message that there is no need for gradients due to resampling.
> - The proof of Theorem 1 can now be found in the appendix.
> - Figures 1 and 2 are simplified and now include DAIS.
>
> >[computation of ESS]
>
> Thank you for this important question. Yes, we compute the ESS from the weights *before* resampling.
>
> >["No Resampling" low at start]
>
> Thank you for this important question. It was hard to see in Figure 1 that the ESS at step 0 is indeed very similar for all methods. We added lines to connect the markers.
>
> >[Why is Theorem 1 not trivially correct?]
>
> Thank you for this comment. We believe that the main technical difficulty of the proof of Theorem 1 is already alleviated by our strict choice of notation. The main technical reason why the proof of Theorem 1 works is that, only if ESS=N, the particle coordinates become independent. Under this condition, we agree that the rest of the proof becomes trivial. But without the independence, Theorem 1 would not be satisfied.
>
> >[drop Theorem 1 to supplement]
>
> Thank you for this important hint. The proof can now be found in the appendix.
>
> >[conditions of Theorem 1 to hold]
>
> The paper now explicitly mentions the caveat in the very first sentence of the discussion.
>
> >[draw parallel to SIXO (Lawson et al., 2022)]
>
> Thank you for pointing us to this really interesting reference. Indeed, this shows an interesting connection to the setup in particle filters. The paper now includes the reference in the discussion section.
>
> >[very condensed and intense technical paper]
>
> Thank you very much for this comment. We now clarified the structure of the paper by introducing subsections for related work and own contributions and moving technical details like the Proof of Theorem 1 to the appendix.
>
> >[variants of the authors work]
>
> Thank you for this comment. Figure 1 and Figure 2 are now simplified and indicate clearly which of the methods utilizes the proposed bound (“DSMCS”). Additionally, we reduced the dependency on acronyms as much as possible.
> Indeed, in our experiments “No Resampling” and “DAIS” show very low effective sample sizes. The paper now improves on the names of the variants, including “ours” whenever the proposed bound is used.
>
> >[Figure 2 is confusing and should be thinned out]
>
> Thank you for this comment. We recognize that Figure 2 was confusing, and we simplified the figure by removing one column. We also clarified the discussion of the figure (first paragraph on page 2), which now walks the reader step by step from DAIS to the proposed method.
>
> >My main request is “thinning” the results figures, and highlighting just a single positive result in the main text. The more complete results can then be deferred to the supplement.
>
> Thank you very much for this comment. As indicated above the Paper now improves on the results being too densely presented and not explained well enough. The paper deferred the theoretical results almost completely to the supplement. Additionally, the paper only mentions one set of models pushing details and additional results to the appendix.
>
> >With this distillation of the key result, drawing explicit connections to Lawson et al., and clarifying how ESS is computed/expanding on the weaknesses of ESS as a metric, I would be comfortable to rating this paper even higher.
>
> Thank you very much for mentioning the possibility of rating this paper higher. We believe that the paragraphs “Related Work” and “Contributions”, the reference of SIXO, and moving much of the technical details to the appendix based on your suggestions helped significantly to improve the paper. If these changes resolve your comments, we would appreciate such an updated rating very much.

---

### Official Review · Reviewer_AyLv · 2023-04-03

**Confidence:** 4

**Summary Of Contributions:**

The authors propose a differentiable variant of sequential Monte Carlo sampling for estimating normalisation constants of sampled distributions. They systematically compare Langevin and Hamiltonian dynamics with the proposed dSMCS under different resampling variants. They conclude that resampling particle weights but omitting the resampling gradients during differentiation, results in better estimates for the normalisation constants and smaller variance in the gradients.

**Rating:**

High Impact (HI): a submission which meets the reviewing criteria and is predicted to make an impact on the field

**Strengths And Weaknesses:**


Strengths:

- This is very well written paper with well designed figures.

- Touches a rather important topic for sampling.

- Has nice structure with outlining the experimental/numerical results first, and then providing theoretical explanations.


Weakness:

- The article is too dense (reasonably for the size of the paper) and includes a lot of acronyms (also reasonably given the permitted length and the number of variants employed here).

- Authors might not need the DEI initiative to publish...

- It is not entirely clear which methods are contributed by the present paper and which are already in the literature and are only used for comparison.

**Suggested Changes:**

- Please increase size of Figure 3.

- A comparison with DAIS (if not too computationally demanding) may be helpful to compare the resulting ESS and ELBOs.

- An implementation that supports the findings and details the proposed resampling variants would greatly facilitate the understanding of the proposed approach.

---

> ### Author Response · Authors · 2023-04-20
> **Response to Reviewer AyLv**
>
> Thank you very much for your review and the time you invested in going through the paper and improving it. Your comments and suggestions addressed some important issues that the paper benefited from being fixed.
>
> >Strengths:
> >This is very well written paper with well designed figures.
> >Touches a rather important topic for sampling.
> >Has nice structure with outlining the experimental/numerical results first, and then providing theoretical explanations.
> >Thank you very much for your kind words. We are glad that you like the paper.
> >The article is too dense (reasonably for the size of the paper) and includes a lot of acronyms (also reasonably given the permitted length >and the number of variants employed here).
>
> Thank you for your comment. We recognize that the article is written very densely. We now “thinned out” the article and made the main message more clear:
> - The paper now contains a section on “Related Work” where we pulled together all parts that discuss how the paper fits into the current research, and a section “Contributions” that summarizes the novelty.
> - The paper only mentions one set of models (No R., Cat, GST), pushing details and additional results to the appendix.
> - The text explicitly conveys the main message that there is no need for gradients due to resampling.
> - The proof of Theorem 1 can now be found in the appendix.
> - Figures 1 and 2 are simplified and now include DAIS.
> - Additionally, we reduced the dependency on acronyms as much as possible.
>
> >Authors might not need the DEI initiative to publish…
>
> Thank you for your kind comment. This really encourages us to go forward with this kind of research. We chose the DEI initiative as the main author fulfills two criteria. We also think that the new “Tiny Paper” format is ideal for presenting these kinds of small results that can be summarized briefly and that can potentially make other researchers work easier, and which might get buried when presented as part of a longer paper.
>
> >It is not entirely clear which methods are contributed by the present paper and which are already in the literature and are only used for comparison.
>
> Thank you for this comment. The new revision of the paper now includes two explicitly labeled sections for “Related Work” and for “Contributions” on page 1.
>
> In brief, the paper contributes the DSMCS method that builds on an SMC bound that has been proposed in the context of particle filters. The method is made differentiable similar to DAIS method by omitting the Metropolis-Hastings correction term. An advantage of the proposed DSMCS method is that it can use various resampling schemes. Crucially, even when using resampling, we observe and explain why resampling gradients may be ignored.
>
> > - Please increase size of Figure 3.
> > - A comparison with DAIS [...]
> > - [share your implementation]
>
> Thank you for these suggestions. We incorporated them all in the new revision of the paper and added a link to a GitHub repository.
>
> Thank you very much for your kind and thorough review. From our point of view, the paper significantly improved in its quality by considering your comments and suggestions. Please let us know if there is anything else we should improve on or clarify.

---

### Author Response · Authors · 2023-05-30
**Opt-In for Archival**

We wish to opt-in for the archival of this paper.

---

### Comment · Area_Chair_2Uqi · 2023-05-30
**Archiving**

Hi All,

The authors have opted-in for archival of this paper.  They updated the paper in light of reviewer feedback, and as a result the paper is even stronger than the initial submission.  I therefore still endorse archival.

Thanks all, and good work!
AC 2Uqi

---

### Meta-Review · Area_Chair_2Uqi · 2023-04-03

**Recommendation:** Invite to present
**Confidence:** 4

**Metareview:**

The work presented is of high quality, dovetails nicely with recent publications in the field, and provides a novel take.  The paper is is well prepared and in the style of an ICLR paper.  However, the authors have condensed it to the point where the paper itself is more difficult to parse than is necessary, reducing the clarity, especially for novice audiences.  This seems to be the main comment from both reviewers.  If these concerns are resolved (and they appear to be very resolvable) then this is a strong submission, and could have significant impact on the wider field.

**Summary:**

This work studies learning models through differentiable samplers.  Similar to some existing methods, the authors find that resampling gradients can be ignored, while also recovering better inference and model learning.  Some empirical and theoretical analysis are presented.

**Comments And Feedback To The Authors:**

Please take on board the comments of the reviewers.  Much of the technical content can be dropped to the supplement, for those interested.  This will clear more space in the main text to include more qualitative descriptions of the method, discussions of comparable work and comparable methods, description of how the baselines were applied, larger figures etc.

The suggestion of a minimal code example is also very noteworthy, since the method is non-trivial.  This can be included as a link to a public GitHub repo.

**Reason For Not Giving A Higher Recommendation:**

Both reviewers raised concerns that the paper was very dense, to the point of obfuscating the core message.  The paper should be "thinned out", so make it more reader-friendly.

There are also some comparisons to existing work (DAIS) and outstanding queries/comments from the reviewers that should be addressed prior to publication.

If these were remedied prior to publication, then I would be inclined to recommend this as a "Notable" submission.

**Reason For Not Giving A Lower Recommendation:**

The work is timely, seems correct, and is a nice addition to the literature.  Both reviewers appraised the work positively.

---

> ### Author Response · Authors · 2023-04-20
> **Response to Area Chair 2Uqi**
>
> Thank you very much for taking the time to meta-review this paper. The paper underwent some significant changes that significantly improved its quality and clarity. The paper improved on several key points:
>
> - The paper now contains a section on “Related Work” where we pulled together all parts that discuss how the paper fits into the current research, and a section “Contributions” that summarizes the novelty.
> - The paper only mentions one set of models (No R., Cat, GST), pushing details and additional results to the appendix.
> - The text explicitly conveys the main message that there is no need for gradients due to resampling.
> - The proof of Theorem 1 can now be found in the appendix.
> - Figures 1 and 2 are simplified and now include DAIS.
>
> We addressed all concerns and questions raised by the reviewers in corresponding responses. Additionally, we included a section linking to a GitHub repository that provides code to reproduce all experiments.
>
> Thank you for mentioning the possibility of recommending our paper as a "Notable" submission. If our updates meet all concerns, we would obviously appreciate this very much.

---

### Decision · Program_Chairs · 2023-04-07

Invite to present